# Parental Self-Compassion and Child Adjustment: The Mediating Role of Parental Depressive Symptoms

**DOI:** 10.3390/ijerph19095133

**Published:** 2022-04-23

**Authors:** Rebecca Y. M. Cheung, Zechen Li, Audrey Pui Lam Ho

**Affiliations:** 1Centre for Child and Family Science, The Education University of Hong Kong, Hong Kong, China; 2Centre for Psychosocial Health, The Education University of Hong Kong, Hong Kong, China; 3Department of Early Childhood Education, The Education University of Hong Kong, Hong Kong, China; s1136649@s.eduhk.hk; 4Department of Psychology, The Education University of Hong Kong, Hong Kong, China; puilamhpl@gmail.com

**Keywords:** self-compassion, parental depressive symptoms, child adjustment

## Abstract

Previous research suggests that self-compassion is associated with mental health and well-being. However, little has been done to understand the role of self-compassion in the family context. Hence, the present study investigated the associations between parents’ self-compassion, parent’s depressive symptoms, and child adjustment. A total 189 Chinese parents (101 mothers) whose children were 2–8 years old were recruited to complete a questionnaire, including measures of parents’ self-compassion, depressive symptoms, and children’s prosocial behavior, internalizing problems, and externalizing problems. Findings indicated mediation effects, in that parents’ depressive symptoms mediated the association between their self-compassion and child adjustment outcomes, namely children’s internalizing and externalizing problems, after controlling for the effects of monthly family income, child gender, and parent gender. Competing hypothesis suggested that parents’ self-compassion did not moderate between parents’ depressive symptoms and child adjustment outcomes. Hence, the association between parental depressive symptoms and child adjustment was not dependent on the level of parents’ self-compassion. As an implication, researchers and practitioners should be made aware of the benefits of parents’ self-compassion on parents’ mental health and child adjustment.

## 1. Introduction

Depression is one of the most prevalent diagnoses in psychopathology. During the first 12 years of a child’s life, as many as 39% of mothers and 21% of fathers have experienced an episode of depression [1]. Ample evidence suggests that parental depression has a negative effect on children, compromising their adjustment with greater internalizing problems, externalizing problems, and sleep problems [2,3,4,5]. Depression is also related to people’s capacity to be compassionate toward themselves [6]. Given early childhood development sets the stage for children’s later adjustment [7,8], it is crucial to delineate the processes underlying child adjustment early on, especially in relation to parental psychopathology.

According to an integrative model of risk transmission to children of depressed parents [9], children’s exposure to parents’ negative affect, behaviors, and cognitions are key mechanisms underlying the effect of parental depression on child adjustment [3,4]. Importantly, parents’ depressive symptoms are associated with a propensity toward negativity, including feelings of inferiority, self-directed and other-directed criticism, and negative views about themselves as parents [9,10,11]. In a study examining paternal speech patterns [12], depressed fathers were found to exhibit more negativity than did nondepressed fathers, such as generating more helplessness comments, self-focused negativity, and negative and critical utterances. In another study [10], mothers’ depression was associated with greater self-directed criticism, which was further associated with children’s insecure attachment. Depression not only is linked to parents’ increased negativity, but is also linked to a reduced capability to care for themselves through self-compassion.

### 1.1. Self-Compassion and Depression

Self-compassion refers to a self-caring attitude in the face of adversity or any perceived inadequacy [13]. According to Neff [13], self-compassion is indexed by self-kindness rather than self-judgment, common humanity (i.e., an understanding that suffering is common and inevitable) rather than isolation, and mindfulness of present-moment experiences rather than over-identification to specific events or experiences. Previous research suggested that depressed individuals exhibited lower self-compassion than did never-depressed individuals, even after controlling for their depressive symptoms [14]. In a cross-lagged panel analysis, Krieger et al. [15] further distinguished the directionality of effects between self-compassion and depressive symptoms. Specifically, they found that self-compassion predicted depressive symptoms longitudinally, but depressive symptoms did not predict subsequent self-compassion. Indeed, self-compassion is a protective factor of mental health. For instance, numerous studies have shown that self-compassion had a negative effect on depressive symptoms [6,14,16,17]. As a result, various evidence-based intervention programs have included a component of self-compassion to improve mental health ([18]; see also [19], for a meta-analysis of randomized controlled trials involving self-compassion interventions). Nevertheless, most of the studies on self-compassion and depression concerns adults from clinical or non-clinical populations [16,17]. Few studies have situated the role of parents’ self-compassion in the family setting.

### 1.2. Parental Self-Compassion and Child Adjustment

Given a lack of research in this area, the relation between parents’ self-compassion and child adjustment remains unclear. While mindfulness-based interventions for parents have incorporated compassion as one of the components to improve parents’ mental health, reduce parenting stress, and improve child adjustment [20,21,22], few have focused specifically on the role of parents’ self-compassion alone on child outcomes. In a study examining the link between self-compassion and child adjustment, Psychogiou et al. [23] found significant correlations between mothers’ and fathers’ self-compassion and children’s internalizing and externalizing problems. However, only fathers’, but not mothers’, self-compassion was associated with children’s subsequent levels of internalizing problems, after controlling for the effects of baseline internalizing problems and parents’ depressive symptoms. Moreover, the relation became nonsignificant upon controlling for multiple comparisons involving other variables. Given their relatively small sample (*N* = 160) and a 34% attrition over time [23], revisiting the potential relation between parents’ self-compassion and child adjustment is deemed necessary.

### 1.3. The Present Study

Drawing from previous findings, parents’ self-compassion might foster a lower level of parental depression [15]. A lower level of parental depression might, in turn, be related to better child adjustment [2,3,10,11]. Hence, we hypothesized that parents’ self-compassion would statistically predict parents’ depressive symptoms. Parents’ depressive symptoms would, in turn, be associated with child adjustment, including children’s prosocial behavior, internalizing problems, and externalizing problems. In addition, we further hypothesized that parents’ depressive symptoms would mediate the association between parents’ self-compassion and child adjustment outcomes, over and above established covariates of parental depression and child adjustment, including child gender [24,25,26], parent gender [27], and family income [28,29]. That is, parental depression would serve as a mechanism between parents’ self-compassion and child adjustment, over and above potential covariates. 

As little has been done thus far to understand the role of parental self-compassion in the family context, it is important to clarify its role in relation to parental depression and child adjustment. Hence, as a competing hypothesis, self-compassion was tested as a moderator between parental depressive symptoms and child adjustment, over and above established covariates of parental depression and child adjustment, including child gender [24,25,26], parent gender [27], and family income [28,29]. Notably, parents’ ability to be compassionate toward themselves amidst the presence of depression might attenuate the negative effect of depressive symptoms on child adjustment. Supporting this hypothesis, previous research showed that caregivers’ positive behavior, such as high quality child care, buffered against the negative effect of parental depression on child adjustment [30]. In another study, mother–child attachment security moderated between maternal depressive symptoms and children’s prosocial behavior in early childhood [31]. Based on these findings, parents’ self-compassion might buffer the association between parental depressive symptoms and child adjustment. That is, as an attribute, parents’ self-compassion might reduce the negative effect of parental depression on child adjustment.

In sum, to illuminate the role of parental self-compassion in relation to parental depression and child adjustment, in this study we first hypothesized a mediation model, in that parents’ depressive symptoms would mediate between parents’ self-compassion and child adjustment. Second, we hypothesized a competing moderation model, such that parents’ self-compassion would reduce the negative effect of parental depression on child adjustment.

## 2. Materials and Methods

### 2.1. Participants and Procedure

The sample consisted of 189 Chinese parents (101 mothers; 88 fathers) whose children were 2–8 years old (*M* = 4.82, *SD* = 2.43). Parents were recruited from flyers at kindergartens and online platforms in major cities of Southern China. Informed consent was sought prior to completion. The average family income per month was RMB33,406.31 (approximately US5284.89). In terms of education, 8.99% completed junior high school or below, 13.25% completed high school, 26.98% had a higher diploma, 40.74% had an undergraduate degree, and 10.04% had a graduate degree. As an incentive, participants were either entered to a lottery or received a supermarket coupon at RMB80 (approximately US$12.59).

### 2.2. Measures

Parents’ Self-compassion. The 12-item Self-compassion Scale Short Form (SCS-SF) [32] was used to assess parents’ self-compassion on a scale from 1 (*almost never*) to 5 (*almost always*). Sample items included, “I try to see my failings as part of the human condition” and “When I feel inadequate in some way, I try to remind myself that feelings of inadequacy are shared by most people.” The raw scores were averaged, with higher scores indicating more self-compassion. In this sample, the Cronbach’s alpha of the sample was 0.62.

Parents’ Depressive Symptoms. The 9-item Patient Health Questionnaire (PHQ-90) [33] was used to measure parents’ depressive symptoms over the past 14 days on a 4-point scale from 0 (*not at all*) to 3 (*nearly every day*). Sample items included, “little interest or pleasure in doing things” and “difficulty for falling asleep.” The raw scores were averaged, with higher scores indicating more severe depressive symptoms. The Cronbach’s alpha of this sample was 0.89.

Children’s Prosocial Behavior, Internalizing Problems, and Externalizing problems. The 25-item Strengths and Difficulties Questionnaire (SDQ) [34] was used to assess children’s prosocial behavior, internalizing problems, and externalizing problems on a 4-point scale from 0 (*not at all*) to 3 (*very much*). Sample items included, “Often offers to help others (parents, teachers, other children)” (prosocial behavior), “Often unhappy, depressed or tearful” (internalizing problems), and “Often loses temper” (externalizing problems). The Cronbach’s alpha of this sample was 0.73, 0.61, and 0.68 for prosocial behavior, internalizing problems, and externalizing problems, respectively.

### 2.3. Analytic Strategies

Zero-order correlations, means, and standard deviations were conducted as preliminary analyses. Path analysis was then conducted via MPLUS, Version 8.3 [35] to investigate the mediating versus moderating models of self-compassion, depressive symptoms, and child adjustment outcomes, with monthly family income, parent gender, and child gender included as covariates. In the mediation model, bootstrapping was used to further investigate the mediation effects. Full information maximum likelihood estimation was used to handle any missing data on the item level.

## 3. Results

Table 1 shows the means, standard deviations, and correlations among the variables. Paired-sample *t*-tests showed that mothers and fathers did not differ on the levels of self-compassion, depressive symptoms, and perceptions of child adjustment (*p*s > 0.05).

### 3.1. Parents’ Depressive Symptoms as a Mediator

The path model fits adequately to the data, χ^2^(2) = 1.33, *p* > 0.05, CFI = 1.00, TLI = 1.06, RMSEA = 0.00, SRMR = 0.01. Parents’ self-compassion was negatively related to their depressive symptoms (β = −0.20, *p* < 0.01), as well as children’s prosocial behavior (β = 0.29, *p* < 0.001), internalizing problems (β = −0.27, *p* < 0.001), and externalizing problems (β = −0.25, *p* < 0.001). Parents’ depressive symptoms, in turn, were related to children’s internalizing problems (β = 0.17, *p* = 0.01) and externalizing problems (β = 0.16, *p* < 0.05), but not their prosocial behavior (β = −0.05, *p* > 0.05), after controlling for the effects of child and parent gender, and monthly family income (see Table 2 and Figure 1 for details).

Based on 5000 bootstrap samples with replacement, the 95% confidence interval (CI) indicated that the standardized indirect effects between parents’ self-compassion and children’s externalizing problems and internalizing problems did not include zeros [CI_internalizing problems_: (−0.11, −0.001) and CI_externalizing problems_: (−0.09, −0.004)]. Therefore, parents’ depressive symptoms mediated between their self-compassion and children’s internalizing problems and externalizing problems.

### 3.2. Alternative Hypothesis: Moderating Effect between Parents’ Self-Compassion and Depressive Symptoms

The path model fits adequately to the data, χ^2^(2) = 1.35, *p* > 0.05, CFI = 1.00, TLI = 1.06, RMSEA = 0.06, SRMR = 0.01. After controlling for the effects of monthly family income and child and parent gender, only the main effect of parents’ self-compassion on children’s prosocial behavior (β = 0.31, *p* < 0.001), internalizing problems (β = −23, *p* < 0.01), and externalizing problems (β = −23, *p* < 0.01) were significant. The main effect of depressive symptoms and the interaction effect between parents’ depressive symptoms and self-compassion were not significant (see Figure 2). Hence, the moderation hypothesis was not supported.

## 4. Discussion

Drawing from the previous literature [10,17,23], the present study investigated the role of parents’ self-compassion in relation to their depressive symptoms and child adjustment. To this end, competing mediation vs. moderation models were tested. Our findings supported the mediation model, in that parents’ depressive symptoms mediated the association between self-compassion and child adjustment outcomes, namely children’s internalizing and externalizing problems, after controlling for previously established effects of monthly family income, child gender, and parent gender. However, parents’ self-compassion did not buffer against the negative effects of parents’ depressive symptoms on child adjustment outcomes. Hence, the association between parental depressive symptoms and child adjustment was not dependent on the level of parents’ self-compassion. Altogether, these findings add to the literature by illuminating the role of parents’ self-compassion in relation to parents’ depressive symptoms and children’s behavioral problems.

Consistent with previous findings on the link between self-compassion and depression in clinical, community, and college student samples [6,15], in this study we found that self-compassion was negatively associated with depressive symptoms among parents. When parents exhibited greater self-kindness, common humanity, and mindfulness of present-moment experiences, they are less likely to be depressed. The present findings further indicated that parental depressive symptoms mediated between parents’ self-compassion and children’s emotional and behavioral problems, thereby suggesting depressive symptoms as a viable mechanism for the link between parental self-compassion and child adjustment. 

Our findings also corroborated previous research on the association between parents’ depressive symptoms and children’s internalizing and externalizing problems [2,3,5,36]. Contrary to previous studies [37,38], however, parents’ depressive symptoms were not related to children’s prosocial behavior, after controlling for the effects of parents’ self-compassion and other demographic variables. The null findings in the path analysis were consistent with the non-significant zero-order correlation between parents’ depressive symptoms and prosocial behavior, as presented in Table 1. Perhaps parents’ depressive symptoms were more robust in predicting children’s negative behaviors such as behavioral problems, rather than positive adjustment such as prosocial behavior. Given our sample involved children at 2–8 years of age, perhaps the children were too young to exhibit a range of prosocial behavior. Indeed, previous research did suggest that prosocial behavior was more likely to occur among older children [39]. Therefore, future studies should further replicate the relation between parental depression and children’s positive adjustment, particularly through a longitudinal approach to capture the trajectories of adjustment from early childhood to adolescence.

In testing the competing moderation hypothesis, parents’ self-compassion did not buffer against the negative effects of parental depressive symptoms on child adjustment outcomes. Contrary to previous research showing that caregivers’ positive behavior buffered against the negative effect of parental depression on child adjustment [30], the protective role of parental self-compassion did not bear out in the moderation model. The null findings may be due, in part, to multicollinearity involving correlated independent variables of parental self-compassion and depressive symptoms (see also Table 1).

## 5. Conclusions

### Limitations and Future Directions

This study has numerous strengths, including an examination of competing hypotheses, the inclusion of both mothers and fathers as participants, and the study of processes between parental depression and child adjustment in an East Asian context, which is lacking in the literature. Despite the strengths, the findings should be interpreted in light of several limitations. First of all, this study used self-report of parents’ depressive symptoms and self-compassion and parent-report of child adjustment. That is, we examined the associations between parent characteristics and parents’ perceptions of children’s outcomes. While parental perceptions of child adjustment are important, future research should incorporate multiple reporters and multiple methods of data collection, such as observations, diary data, and physiological data, to reduce self-report biases. Second, the present study had a cross-sectional design. To draw conclusions on the temporal sequence and directionality of effects, longitudinal studies are necessary. Third, the internal consistency of the measures for parents’ self-compassion, children’s internalizing problems, and children’s externalizing problems were low, with Cronbach’s alphas below 0.70. Previous research suggested that the magnitude of Cronbach’s alpha can be affected by the length of a scale, with scales over 20 items typically achieving acceptable alpha (e.g., above 0.70) [40]. In the present study, parents’ self-compassion, children’s internalizing problems, and children’s externalizing problems had 12 items, 10 items, and 10 items, respectively. Although other studies involving Chinese samples similarly reported low alphas for scales of self-compassion [41] and parent-reported internalizing and externalizing problems [3], the present findings should be interpreted with caution.

Notwithstanding the limitations, the present study lends support to the mediation effects of parents’ depressive symptoms between parents’ self-compassion and child adjustment outcomes. The findings also refuted the moderation hypothesis, i.e., parents’ self-compassion did not buffer against the negative effects of parents’ depressive symptoms on child adjustment outcomes. As a major implication, researchers and practitioners should be made aware of the protective role of parents’ self-compassion on parents’ mental health and child adjustment. Along the same lines, researchers and practitioners could develop evidence-based interventions involving self-compassion to directly reduce parental depressive symptoms and indirectly enhance child adjustment. The present findings also inform theories of self-compassion [13] by showing its relevance to the family context. To understand the effects of parental self-compassion and depression on children across developmental periods, longitudinal studies merit future investigations.

## Figures and Tables

**Figure 1 ijerph-19-05133-f001:**
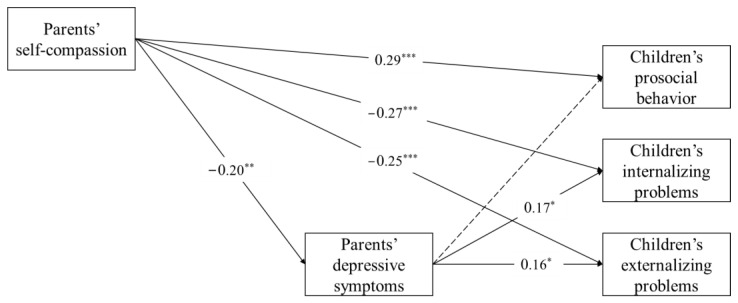
Final model of depressive symptoms as a mediator between parents’ self-compassion and child adjustment. χ^2^(2) = 1.33, *p* > 0.05, CFI = 1.00, TLI = 1.06, RMSEA = 0.00, SRMR = 0.01. Covariates including monthly family income and child and parent gender were included as control variables but are not depicted in the figure for clarity. Non-significant paths are shown in dashed arrows. * *p* < 0.05, ** *p* < 0.01, *** *p* < 0.001.

**Figure 2 ijerph-19-05133-f002:**
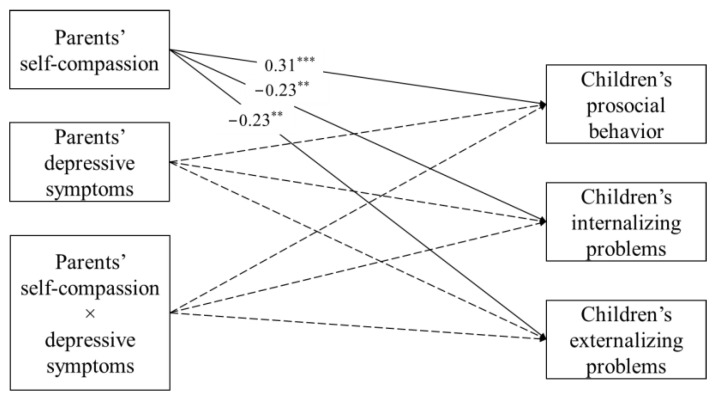
Final moderation model of parents’ self-compassion and depressive symptoms on child adjustment. χ^2^(2) = 1.35, *p* > 0.05, CFI = 1.00, TLI = 1.06, RMSEA = 0.00, SRMR = 0.01. Covariates including monthly family income and child and parent gender were included as control variables but are not depicted in the figure for clarity. Non-significant paths are shown in dashed arrows. ** *p* < 0.01, *** *p* < 0.001.

**Table 1 ijerph-19-05133-t001:** Means, standard deviations, and zero-order correlations of the variables under study.

Variable	*M*	*SD*	(1)	(2)	(3)	(4)	(5)	(6)	(7)
(1) Child gender	—	—	—						
(2) Parent gender	—	—	0.00	—					
(3) Monthly family income (in RMB)	33,406.31	31,861.49	0.10	−0.41 ***	—				
(4) Parents’ self-compassion	3.18	0.46	−0.01	0.03	0.10	—			
(5) Parents’ depressive symptoms	0.29	0.51	0.00	−0.01	−0.02	−0.21 **	—		
(6) Children’s prosocial behavior	2.24	0.61	0.16 *	−0.00	0.10	0.31 ***	−0.12	—	
(7) Children’s externalizing problems	1.51	0.28	0.14	0.02	−0.22 **	−0.33 ***	0.26 ***	−0.20 **	—
(8) Children’s internalizing problems	1.76	0.31	−0.03	0.07	−0.13	−0.30 ***	0.23 **	−0.24 ***	0.49 ***

Note. * *p* < 0.05, ** *p* < 0.01, *** *p* < 0.001.

**Table 2 ijerph-19-05133-t002:** Parameter estimates of the model with parents’ depressive symptoms as a mediator.

Parameter	Unstandardized *B* (*SE*)	Standardized β
Parents’ self-compassion →		
Parents’ depressive symptoms	−0.21 (0.08)	−0.20 **
Children’s prosocial behavior	0.37 (0.09)	0.29 ***
Children’s internalizing problems	−0.16 (0.04)	−0.27 ***
Children’s externalizing problems	−0.17 (0.05)	−0.25 ***
Parents’ depressive symptoms →		
Children’s prosocial behavior	−0.06 (0.08)	−0.05
Children’s internalizing problems	0.09 (0.04)	0.17 *
Children’s externalizing problems	0.10 (0.04)	0.16 *
Child gender →		
Parents’ depressive symptoms	0.01 (0.07)	0.01
Children’s prosocial behavior	0.19 (0.08)	0.16 *
Children’s internalizing problems	0.09 (0.04)	0.15 *
Children’s externalizing problems	−0.01 (0.04)	−0.02
Parent gender →		
Parents’ depressive symptoms	−0.04 (0.08)	−0.04
Children’s prosocial behavior	0.03 (0.09)	0.02
Children’s internalizing problems	−0.02 (0.04)	−0.04
Children’s externalizing problems	0.02 (0.04)	0.04
Monthly family income →		
Parents’ depressive symptoms	−0.05 (0.03)	−0.13
Children’s prosocial behavior	0.06 (0.04)	0.12
Children’s internalizing problems	−0.05 (0.02)	−0.23 **
Children’s externalizing problems	−0.03 (0.02)	−0.12
Parents’ self-compassion ←→		
Child gender	−0.00 (0.02)	−0.02
Parent gender	0.01 (0.02)	0.03
Family income	0.06 (0.04)	0.11

Note: * *p <* 0.05, ** *p* < 0.01, *** *p* < 0.001.

## Data Availability

The data presented in this study are available on request from the corresponding author, R.Y.M.C. The data are not publicly available due to ethical restrictions.

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
