# Peer review of "Parental Self-Compassion and Child Adjustment: The Mediating Role of Parental Depressive Symptoms"

_ijerph, 2022, doi:10.3390/ijerph19095133_

Round 1

Reviewer 1 Report

Thank you for the opportunity to review the manuscript entitled "Parental Self-Compassion and Child Adjustment: The Mediating Role of Parental Depressive Symptoms."  In this manuscript, the authors explore the relationships between parental depression, parental self-compassion, and child outcomes including prosocial behavior and internalizing and externalizing problems.  The authors find that parental depression mediates the link between parental self-compassion and child outcomes, and suggest that parental self-compassion should be further explored as a protective factor.

The manuscript is well written and I do not have any minor edits to point out.  My concerns lie with the limitations that the authors acknowledged, and I think they should elaborate more on these points in the discussion:

  • As the authors mentioned, the Cronbach's alphas for many of the measures are low.  Can they provide citations of other studies that have been published with similarly low reliability of measures?
  • Also as the authors mentioned, each of the variables are assessed via parent report.  In particular, parents rated their children's own outcomes (prosocial behavior, internalizing and externalizing problems).  So really the associations are between parent characteristics (self compassion and depression) and THEIR PERCEPTIONS of children's outcomes.  This is not necessarily a limitation, as parental perceptions of children's adjustment is important in and of itself.  But the authors should elaborate on this point.

Reviewer 2 Report

Line 32. add "and" before sleep problem.

Line 36, 37, unclear

The second paragraph of the introduction needs to be developed more to support the association between depression and child adjustment, proposed in the study.

In the second and third paragraphs of the Introduction, more clearly describe what the previous studies mean to this study. 

Line 70 change “.” to “,”

Add more explanations of the child adjustment in the Introduction and why understanding the child adjustment is important since it is the dependent variable of the study.

Plus overall, the literature review in the Introduction needs to be developed more.

The authors have family income and parent and child gender as covariates. The authors need to more clearly explain in the introduction why these variables are necessary to be included in this study.

The authors need to thoroughly talk about the need and justification for the moderating effect hypothesis in the Introduction. Presenting results of the analysis of the moderating model cannot justify and support the proposed mediating model.

The authors conducted mediating and moderating analyses together. The authors need to describe the difference of the two analyses, what’s the strengths and weaknesses of the two approaches, and why this study needs to use the two approaches together. Presenting the moderating analysis can’t make the proposed mediating model sound without theoretical and conceptual supports.  

A Figure is needed for the moderating analysis.

Line 148, 149. The values of the model fit need to be double-checked. CFI 1.00 and RMSEA .00 ??

Line 190, 191, only with the statistical analysis, we can’t propose that. Please provide more explanations.

 In the discussion, the authors need to provide more theoretical and practical implications of the study. Without using the competing hypothesis, which is not a huge plus for this study, it is difficult to understand the theoretical and practical contributions of the study.

Reviewer 3 Report

The authors seek to extend findings related self-compassion as a protective factor against depression in a large sample of Chinese mothers and fathers -- going further as to explore the effects of parental self-compassion on their children's adjustment and externalizing and internalizing symptoms.  The paper, while adequate from a methodological/analysis perspective, suffers from some curious analytic choices and unclear writing; a good amount of English language checking is in order.  I offer some key examples below:

  1. Paragraph 1, lines 32-35 aren't saying anything: "developmental psychopathology perspective highlights multiple pathways toward child development".  What pathways?  Why is it important to understanding pediatric depression?

  2. P. 2, ln 46-47 (topic sentence) directly contradicts the findings reported in the rest of the paragraph.  Lines 65-66 would actually be a better topic sentence.

  3. Lines 67-81 (p. 2, paragraph 2) reports a series of slightly different findings regarding self-compassion without coming to a point.

  4. p. 2, lines 93-95 are redundant with the the previous two sentences
    P. 3, lines 93-98 are also unnecessary.

Regarding methodology...The Cronbach's alphas for all of the measures except parental depression are quite weak (.62 for self-compassion and .61-.73 for child psychopathology and adjustment), bringing the whole path analysis into question.  How reliable are these findings really.  These weak internal consistencies should be cited as a limitation.  I also recommend that Table 2 precede Figure 1 for greater clarity, and I ask the authors to explicate their reasoning for including their chosen covariates.  (I assume there is literature justifying them; but Table 2 would suggest that parent gender need not have been included in the model at all and may have actually weakened your results).  

Round 2

Reviewer 2 Report

Authors responded to comments. The current format of the manuscript is ready to be published after minor editing.

Reviewer 3 Report

Thank you for your close attention to our suggestions.  The manuscript is significantly improved and reads much more clearly with consistent messaging.  Well done.